# A Pilot Study on the Prediction of Non-Contact Muscle Injuries Based on *ACTN3* R577X and *ACE* I/D Polymorphisms in Professional Soccer Athletes

**DOI:** 10.3390/genes13112009

**Published:** 2022-11-02

**Authors:** Kathleen Y. de Almeida, Tiago Cetolin, Andrea Rita Marrero, Aderbal Silva Aguiar Junior, Pedro Mohr, Naoki Kikuchi

**Affiliations:** 1Graduate School of Health and Sport Science, Nippon Sport Science University, Tokyo 158-8508, Japan; 2Graduate Program in Neurosciences, Federal University of Santa Catarina, Araranguá 88905-120, Brazil; 3Graduate Program in Cell and Developmental Biology, Federal University of Santa Catarina, Florianópolis 88040-900, Brazil; 4Sports Center, Federal University of Santa Catarina, Florianópolis 88040-900, Brazil

**Keywords:** polymorphism, *ACTN3*, *ACE*, soccer, muscle injury

## Abstract

Muscle injuries are among the main reasons for medical leavings of soccer athletes, being a major concern within professional teams and their prevention associated with sport success. Several factors are associated with a greater predisposition to injury, and genetic background is increasingly being investigated. The aim of this study was to analyze whether *ACTN3* R577X and *ACE* I/D polymorphisms are predictors of the incidence and severity of muscle injury in professional soccer athletes from Brazil, individually and in association. Eighty-three professional athletes from the first and second divisions of the Brazilian Championship were evaluated regarding the polymorphisms through blood samples. Nighty-nine muscle injuries were identified during the seasons of 2018, 2019 and 2020 and categorized according to severity. *ACTN3* XX individuals had a higher frequency of severe injuries compared to the RX and RR genotypes (*p* = 0.001), and in the dominant model (compared to RX+RR), with *p* < 0.001. The trend *p*-value test showed an increased number of injuries/season following the order XX > RX > RR (*p* = 0.045). Those with the *ACE* II genotype had almost 2 fold the number of injuries per season compared to those with the ID+DD genotypes (*p* = 0.03). Logistic regression showed that the polymorphisms are predictors of the development of severe injury (*ACTN3* R577X model with *p* = 0.004, R^2^: 0.259; *ACE* I/D model with *p* = 0.045, R^2^: 0.163), where *ACTN3* XX individuals were more likely to suffer from severe injury (OR: 5.141, 95% CI: 1.472–17.961, *p* = 0.010). The combination of the *ACTN3* 577X allele and the *ACE* II genotype showed an increased number of injuries per season, enhanced by 100% (1.682 injuries/season versus 0.868 injuries/season, *p* = 0.016). Our findings suggest that both polymorphisms *ACTN3* R577X and *ACE* I/D (and their interaction) are associated with the susceptibility and severity of non-contact muscle injury in soccer players.

## 1. Introduction

Elite performance in soccer is often determined by the execution of high-intensity running, which corresponds to running at moderate speed, high speed, and sprinting [1,2]. These moments can be decisive for the game results since the ability to perform faster muscular contractions can be a performance-limiting factor [3] and a way of separating high-level athletes from athletes of a lower standard [4]. The multiple accelerations and decelerations involved in soccer require the application of concentric and eccentric muscle contractions [5], in which the latter occurs when the muscle is stretched, hence, it is generally a crucial risk factor in muscular injuries in most sports, and especially in soccer [5,6].

The risk of injury in soccer is approximately 1000-fold higher than that in other sports [7,8]. For instance, muscle injuries have been the primary cause for retirement of professional soccer athletes [7], accounting for 37% of all loss injuries to professional male soccer athletes [9]. Simultaneously, it has also been linked to success in the sport especially of soccer teams, as demonstrated by a study of Qatari professional soccer teams, in which it was found that lower incidence rates were directly related to higher team ranks owing to the higher number of victories and team goals [10]. Several intrinsic factors such as age [11], length of career [11], lack of flexibility [12], decline or imbalances in muscle strength [13], presence of previous injuries [13,14], and genetics determine the risk of athletes to muscle injuries [8].

Among the genetic factors associated with muscle injury, the *ACTN3* R577X polymorphism (rs1815739) is the most common. It is characterized by the single-nucleotide polymorphism (SNP), where an exchange of cytosine for thymine occurs, resulting in the replacement of the amino acid arginine to a premature stop codon in the protein α-actin-3 site located in the Z line of the sarcomere [15]. In this polymorphism, higher strength and power are associated with the R allele, while better muscle endurance is associated with the mutated allele, called the X allele [16,17].

Additionally, the insertion (I) or deletion (D) of the *ACE* gene (rs4646994) is a polymorphism related to lesion development, in which it encodes the angiotensin-converting enzyme (*ACE*) [18], or the enzyme involved in the renin–angiotensin–aldosterone system that controls blood pressure levels and regulates homeostasis. In this polymorphism, the D allele, which lacks a 287 bp sequence in intron 16, is associated with higher *ACE* activity in tissues [19]. Homozygotes of the D allele have been found to have a higher enzyme activity compared to the ID and II genotypes, and the presence of the D allele is related to muscular strength and found in higher frequency in high-speed and high-performance sports athletes [20,21], while the presence of the I allele is associated with endurance sports athletes [22,23].

The type of ACTN protein produced directly influences the markers of muscle damage and inflammatory responses associated with exercise and sport [23,24], and for being directly related to muscle structure its alteration reflects on the muscle fiber characteristics [25]. The absence of the *ACTN3* protein in soccer athletes was found to decrease the clearance rate of creatine kinase (CK), which is a blood marker of sarcolemma disruption associated with muscle damage [26]. Similarly, the I allele of the *ACE* I/D polymorphism is linked to a higher presence of markers of post-exercise muscle damage [27].

Both polymorphisms have been previously associated with higher injury incidence and severity [28,29,30]. However, information on the relationship between genetic factors and muscle injury susceptibility has remained lacking [31]. Findings from previous studies have also been inconsistent, with differences between populations, lack of association [31,32,33,34,35], and the effect of the polymorphisms can vary between untrained individuals and professional level, as pointed out by a mini-review in the *ACTN3* polymorphism [25]. Moreover, the methods used have been different, varying population sizes [30], independent candidate gene associations, and no interaction analyses have been used [31].

Hence, this study aims to analyze the association between *ACTN3* R577X and *ACE* I/D polymorphisms and the development of non-contact muscle injuries, including their incidence and severity in high-performing soccer male athletes from Brazil. This study also determined both isolated candidate polymorphisms and their interactions.

## 2. Materials and Methods

### 2.1. Cohort and Data Collection

Eighty-three male professional athletes belonging to three professional teams from first and second Division of the Brazilian Championship (Brasileirão) participated in this study. Inclusion criteria for the group of athletes were: male professional soccer athletes, belonging to professional soccer clubs from the 1st and 2nd divisions of the Brazilian Soccer League, with a minimum of 2 uninterrupted years of sports practice before the beginning of this study. On the other hand, the exclusion criteria for the group of athletes were: bone and/or muscle injuries in the three months prior to the beginning of this study and failure to meet the inclusion criteria.

A total of ninety-nine non-contact muscle injury occurrences were identified and diagnosed by the medical group of each soccer team during the seasons of 2018, 2019 and 2020 seasons. Injuries were categorized according to Fuller and collaborators [36], dividing the severity according to days of medical leaves, wherein in the present study the “severe” group was made up of injuries that caused more than 28 days of absence and the “non-severe” group included all the injuries that caused less or equal to 28 absence days.

We collected information on age, weight, and height of each athlete, from the records of each soccer club. Further, to analyze if years of experience (career time) is a factor that can influence predisposition to severe injury, the history of each player was raised, counting the years of “experience” from the moment they started playing as professionals (excluding youth categories) until the year in which the injury occurred. This information was assessed through the oGol website (www.ogol.com.br accessed on 1 August 2022), which is among the world’s largest open-access football databases.

This study was approved by the Ethics Committee on Human Research at the Federal University of Santa Catarina under number 3,621,353/2019. Participation was conditional based on (i) voluntary participation of the subjects, and (ii) informed consent. This research was conducted in accordance with Resolution 466/2012 of the Brazilian National Health Council.

### 2.2. DNA Analyses

At the beginning of the season, the athletes participated in a routine collection of 4 mL blood samples, performed by qualified professionals. Blood was collected in BD Vacutainer^®^ tubes and DNA extraction was performed with the salting out protocol: 100 μL of the obtained leukocyte layer was resuspended in lysis solution (0.1 M Tris-HCl, 0.32 M sucrose, 0.0025 M MgCl2, 10 mL Triton X 100 1%, pH 7.6), 300 μL of a second lysis solution (0.1 M Tris-HCL, 0.05 M KCl, NONIDET p-40 1%–Stigma 500 mL–Tween 20 1%–VETEC 1 L, pH 8.5), 10 μL of 10% SDS and 75 μL of 5 M sodium perchlorate, 130 μL of saturated 6 M NaCl, 300 μL of isopropyl alcohol, 300 μL of 70% ethanol and 200 μL of MilliQ water and the samples were incubated at 56 °C in a water bath for 30 min. Samples were stored in an overnight refrigerator for further storage at −20 °C. The DNA samples obtained were quantified as to their purity in a Nanovue Plus^®^ spectrophotometer (Eppendorf) and diluted by adding MilliQ Water to 50 ng of DNA per μL.

Genotyping for *ACTN3* R577X was performed by polymerase chain reaction (PCR) and restriction fragment length polymorphism with a restriction enzyme (RFLP). The protocol was adapted from Mills and colleagues [37]. Exon 16 of the *ACTN3* gene (rs1815739) was amplified with the primer sequence, anchored to adjacent intronic sequences: direct 5′-CTGTTGCCTGTGGTAAGTGGG-3′ and reverse 5′-TGGTCACAGTATGCAGGAGGG-3′. The PCR product was subjected to enzymatic digestion with 10 U of Ddel enzyme (New England BioLabs) according to the fabricant instructions. The fragments were separated by electrophoresis on 3% agarose gel.

Genotyping for *ACE* I/D was performed according to a protocol adapted from Moraes and collaborators [38]. Intron 16 of the *ACE* gene (rs4646994) was amplified with the primer sequence: direct 5′-CTGGAGAGAGCCACTCCCATCCTTCT-3′ and reverse 5′-GAYGTGGCCATCACATTCGTCAGAT-3′. The fragments were separated by electrophoresis on 1% agarose gel. Shanmugan et al. [39] identified a preferential amplification by the deletion allele D to the detriment of allele I. For this reason, all homozygotes for D were re-evaluated through a new PCR to identify possible Insertion alleles that were not amplified in the first step, modifying the direct primer to bind on the insertion fragment: direct 5′-TTTGAGACGGAGTCTCGCTC-3′ and reverse 5′-GAYGTGGCCATCACATTCGTCAGAT-3′.

### 2.3. Statistical Analyses

Statistical analyses were performed using SPSS statistical package version 25.0 for Windows (SPSS Inc., Chicago, IL, USA). The Hardy–Weinberg equilibrium for both genotypes was accessed using Pearson’s χ2 test. Comparisons between genotypes and severity groups (severe versus non-severe), and injury per season were made using the χ2 test and Fisher’s Exact Test. SNP stats (https://snpstats.net accessed on 24 October 2022) was used to verify the Akaike Information Criterion (AIC) for the most suitable genetic model. Binary logistic regression was performed to verify whether the *ACTN3* R577X genotypes and the *ACE* I/D recessive model are predictors of the development of severe injury, creating a model and calculating the odds ratio (OR) and 95% confidence intervals (CI). The interaction between the polymorphisms and injury per season was assessed with Analysis of Covariance (ANCOVA) adjusted by weight and age. *p*-values < 0.05 were considered statistically significant.

## 3. Results

The frequencies in both polymorphisms did not deviate from the Hardy–Weinberg equilibrium (*ACTN3* R577X *p* = 0.95 and *ACE* I/D *p* = 0.2). Table 1 shows subject characteristics according to the *ACTN3* and *ACE* genotypes. Individuals with the *ACTN3* XX genotype were significantly older than those with the RX and RR genotypes (*p* = 0.030), and they had a tendency towards more experience years at the professional level (*p* = 0.079). For *ACE* I/D polymorphism, those with the II genotype had a tendency of having more years at the professional level of soccer (*p* = 0.059).

Injured subjects’ characteristics are shown in Table 2, divided into those who had severe and those who had non-severe injuries. The group of “non-severe” injuries presented significantly higher weight compared to the group of “severe” injuries (*p* = 0.032).

As shown in Table 3, for *ACTN3*, there was an association between severity and genotype, and also for the dominant model, where XX individuals had a higher frequency of severe injuries compared to those with the RX and RR genotypes with *p* = 0.001 (Figure 1), and compared to those with RX + RR (dominant model), with *p* < 0.001. The *ACTN3* polymorphism also had a trend (*p* = 0.079) for the number of injuries per season in the recessive model, where individuals containing at least one X allele (RX + XX) had a double frequency of injuries per season and the trend *p*-value test was significant (*p* = 0.045), where the genotypes increased the number of injuries per season following the order XX > RX > RR (Figure 2).

For the *ACE* I/D polymorphism, as shown in Table 4, the dominant model showed an association with the number of injuries per season, where individuals with the II genotype had almost 2 fold the number of injuries per season compared to those with the ID + DD genotypes with *p* = 0.03. There was also a trend in the recessive model, where the DD genotype had more severe injuries than the II + ID genotypes (*p* = 0.063).

AIC pointed out that the dominant model of *ACTN3* and the recessive model of *ACE* are the most suitable models for analysis of severity (AIC 57.2, *p* = 7 × 10^−4^; AIC 64.6, *p* = 0.04, respectively). Therefore, a logistic regression analysis was performed to verify whether the polymorphisms are predictors of the development of severe injury (severe injury vs. non-severe injury) (Table 5). The model containing the *ACTN3* R577X genotypes and using the XX genotype as a predictor was significant (*p* = 0.004, R^2^: 0.259). The model showed no association between weight and years of experience (*p* > 0.05) with injury severity but there was an association for *ACTN3* polymorphism, where XX individuals are more likely to suffer from severe injury (OR: 5.141, 95% CI: 1.472–17.961, *p* = 0.010). The model containing the *ACE* I/D polymorphism in the recessive model using the DD genotype as a predictor was also significant (*p* = 0.045, R^2^: 0.163), but there is only a trend toward the *ACE* genotypes, where the DD genotype is more associated with severe injury compared to the II + ID genotypes (OR: 3.437, 95% CI: 0.862–13.701, *p* = 0.080. In this same model, another factor showing a trend was the weight (*p* = 0.066). Finally, Model 3 included both polymorphisms and was also significant (*p* = 0.003, R^2^: 0.315). In this model, individuals with XX of *ACTN3* R577X and II of *ACE* I/D were more likely to suffer from severe injury compared to the other counterparts.

There was an interaction between the *ACTN3* R577X polymorphism and *ACE* I/D. The presence of the *ACTN3* X allele with the *ACE* II genotype increases the number of injuries per season by 100%, meaning that carriers of the combination X+II have double the chance of injury per season, being 1.682 injuries/season, whereas the other subjects (carriers of the combination *ACTN3* RR genotype + *ACE* D allele) have 0.868 injuries per season (*p* = 0.016) (Figure 3). Other combinations did not show any significant relationship (*p* > 0.05).

## 4. Discussion

The results indicate that there is a relationship between the *ACTN3* R577X and *ACE* I/D polymorphisms and injury development in Brazilian soccer players, based on the correlation found between the XX genotype of *ACTN3* R577X and muscle injury severity in the genotype and dominant models when compared with the RX + RR combination. Further, the presence of the X allele (XX + RX) also exhibited a relationship with the number of injuries per season, in which the addition of the X allele was found to significantly increase the risk of injuries in each season. The *ACTN3* R577X genotype was identified to be a significant predictor of injury severity, in which the XX genotype had a 5.14 higher likelihood for severe muscle injury, as compared with other genotypes. For *ACE* I/D, genotype II was observed to be associated with higher injury incidence in the dominant model, as compared with the ID + DD combination. Finally, an interaction was found between the polymorphisms, in which the combination of the X allele of *ACTN3* with *ACE* I/D genotype II resulted in higher injury incidence per season by 2 fold.

Similar studies on the relationship between the *ACTN3* R577X polymorphism and injury risk have shown inconsistent results in the past. For instance, previous studies found no association between *ACTN3* polymorphism and injury incidence in soccer athletes, marathon runners, endurance runners, and university athletes, respectively. Meanwhile, other studies were found to be consistent with the results in the present study [32,33,34,40], in which a study by Clos and collaborators involving 43 soccer players found that the XX genotype obtained the highest injury incidence rate compared with the RR and RX genotypes. Here, no significant association was found with the genotypes of *ACTN3* R577X [28] in terms of the severity and recovery time. Additionally, the results of a study by Massidda and collaborators [29] involving professional soccer players from the Italian first division indicated that the *ACTN3* XX genotype had higher odds for injuries compared with the RR genotype with a 2.66 likelihood, which was consistent with the results in this study that showed a 5.14 likelihood for XX in Brazilian athletes. The study also identified a higher injury severity likelihood for XX individuals with 2.13 higher likelihood compared with the RR counterparts [29].

In female university athletes, the *ACTN3* R577 allele was observed to have a higher frequency among the test subjects without muscle injuries [35]. However, Del Coso and collaborators found no relationship between non-contact injury incidence, injury type, recovery time, and other analyzed parameters with the *ACTN3* polymorphisms [41], indicating that there may be a difference between female and male athletes, hence, requires further investigation.

Vincent et al. [42] suggested that the α-actin-3 protein has a protective effect in muscle injuries. Hence, this may explain the absence of protein (XX) in the genotype, as it had the highest severity of muscle damage and signaling during tissue repair. Several studies have also indicated that the deficiency of α-actinin-3 increases the risk of muscle damage and enhances catabolic response following strenuous exercise. For instance, higher creatine kinase (CK) activities were found to have higher levels of cortisol following acute eccentric trainings [26], in which the XX individuals involved in these studies were 3 fold more likely to develop exertional rhabdomyolysis [43].

In this study, it is inferred that the deficiency of α-actinin-3 may be partially compensated by α-actinin-2, which is a highly homologous protein. However, this may lead to different protein interactions in the muscle, particularly the functional and operating functions, as these are products of gene duplication. Hence, the differences between the XX and RR carriers of the *ACTN3* R577X polymorphism are determined to be products of a compensatory upper regulation of α-actinin-2 in the XX genotype, leading to muscle remodeling. The interactions with cellular proteins such as Z-band alternatively spliced PDZ motif-containing protein (ZASP), titin, myotilin, desmin, and vinculin [44] may also lead to a higher susceptibility of the muscle to skeletal damage induced by the contraction [44,45].

Massidda et al. [30] found a relationship with muscle injury in the *ACE* I/D polymorphism, in which the D allele had a significantly lower frequency in the “injured” subject group in both the Italian and Japanese male soccer players; however, no association between the *ACE* I/D genotypes and injury severity was reported. Similarly, Larruskain et al. [32] also reported no relationship between the *ACE* I/D polymorphism and muscle injury.

Previously, the *ACE* I/D polymorphism was found to be associated with exercise-induced muscle damage based on an analysis of blood circulatory CK levels in triathlon and marathon runners [46,47]. Here, one or two copies of the D allele were found to be associated with lower CK response levels. The polymorphism was also associated with other types of muscle phenotypes related to exercise induction, such as muscle metabolism itself [48], cardiac muscle growth responses [49], fiber distribution, and composition difference [50].

However, the I allele was primarily related to a higher muscle injury incidence owing to its relationship with the inflammatory response, based on the significant relationship found with higher injury incidence in each season. The activity of the angiotensin-converting enzyme (*ACE*) modulates the renin–angiotensin system and the kallikrein–kinin system in the blood. Higher occurrences of *ACE*, as reported for the D allele and the DD genotypes, also suppress the levels of bradykinin, which is a peptide that induces inflammatory processes and extravasation of myocellular proteins [51,52]. This indicates that alleles I and II were related to a higher injury susceptibility, owing to their higher inflammatory responsiveness. The interaction between the two polymorphisms showed, as a novelty, that the combination of the *ACTN3* X allele and the *ACE* II genotype in Brazilian soccer athletes results in a significantly higher incidence of non-contact muscle injury per season. This higher incidence may be attributed to a combination of predisposing factors of muscle injuries that involve risk alleles and genotypes. Among the subjects studied, muscle remodeling and metabolism changes were induced by both polymorphisms that were crucial determinants of a muscle injury. Additionally, both the X allele of *ACTN3* and the I allele of *ACE* were related to less muscle strength, which is a risk factor for injury in itself, was notably an isolated characteristic that was found to have higher severity for subjects that have a combination of this allele and genotype. Another factor that can be taken into consideration is that studies have shown that soccer is becoming an increasingly faster sport. A study in the English Premier League indicated that across seven seasons, high-intensity running distance and actions increased by ~30%, and only happened to be a 2 to 4% increase in the total distance covered, considering the last decade [53,54]; therefore, carriers of the combination of both genotypes that do not confer a better high-speed ability might be being overexerted and, therefore, have a higher injury incidence.

Athletes with the XX genotype were significantly older, and age is a factor that can lead to a higher risk of injury [14,55]. The higher risk of injury in older players is still unclear but can be attributed to increased body weight, for example [56]. However, we did not observe differences in the body weight between the athletes divided into genotypes. Moreover, the regression analysis did not show a relationship with experience years, and the interaction analysis was adjusted by age and weight.

The development of training and tactics based on injury prevention, as well as recovery programs, can be better guided by using inherent information such as the genetic background of athletes. By identifying and understanding the genetic characteristics of athletes and how they can influence their injury predispositions followed by the exercise exposure, the training intensity can be adjusted at certain times of the season, as well as workloads, and methods of training and plan individualized recovery after the injury. Those actions can lead to performance enhancement and prevention of over-demand that causes chronic situations.

Several limitations such as the absence of biomarkers for muscle injury were noted in this study, which may result in micro-injuries that may not necessarily cause the athletes’ absence, but instead can contribute to a long-term medical situation. Further, the volume of training and matches, that could provide a better understanding of the relationship between training load and effort in injury development according to genetic background, were not analyzed. The number of professional athletes studied was also limited, hence, the results should be interpreted with caution and it is recommended that future studies include a higher subject population to verify the results. Additionally, because of the limitation in the number of subjects, differences between athletes from the first and second division were not verified, together with the fact that there was a fluctuation of some of the studied teams during the seasons regarding their classification in the Brazilian Championship (between first and second divisions). Finally, to the best of our knowledge, although other studies have found a relationship between the polymorphisms and the development of muscle injury, this is the first study to determine the relationship between two polymorphisms and the risk of non-contact muscle injury. Therefore, other polymorphisms and their interactions are also recommended to be investigated, considering that gene interaction and several metabolic pathways are involved in numerous genetic processes.

## 5. Conclusions

In conclusion, the findings of this study suggest that the *ACTN3* R577X and *ACE* I/D polymorphisms are genetic variants associated with susceptibility to non-contact muscle injury in male soccer players. The I variant of the *ACE* polymorphism has a higher incidence of muscle injury per season, similar to the X allele of the *ACTN3* polymorphism, which was also found to be related to the development of severe muscle injury, resulting in an extended recovery time for players. The combination of polymorphisms in subjects with the *ACTN3* 577X allele and the *ACE* II genotype is a predisposing factor for the frequency of injuries per season, which was found to be 2-fold higher than that of other combinations. Hence, the results suggest that considering genetic factors is crucial for preventing injuries and consequently improving sports recovery and rehabilitation.

## Figures and Tables

**Figure 1 genes-13-02009-f001:**
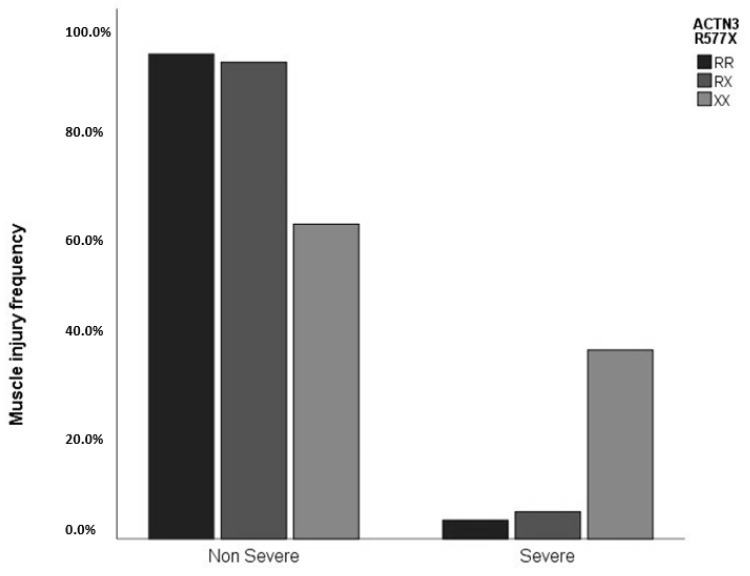
Distribution of severe and non-severe muscle injuries according to the *ACTN3* R577X genotypes (*p* = 0.001).

**Figure 2 genes-13-02009-f002:**
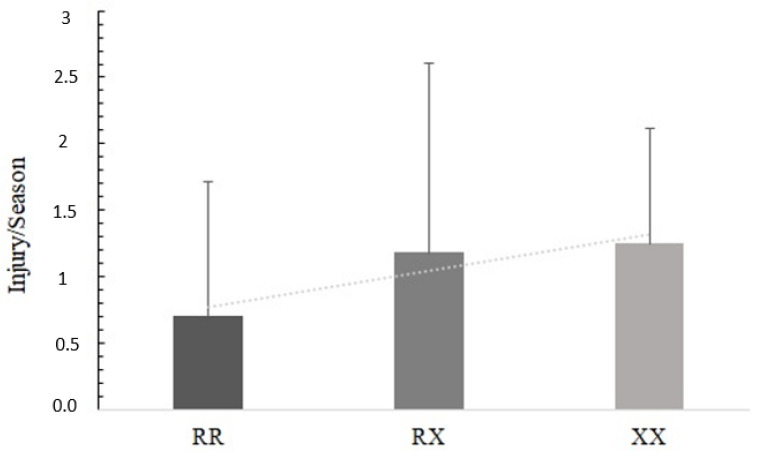
Trend *p*-value of the *ACTN3* R577X genotypes and injury per season. There is an increase following the order XX > RX > RR (*p* = 0.045).

**Figure 3 genes-13-02009-f003:**
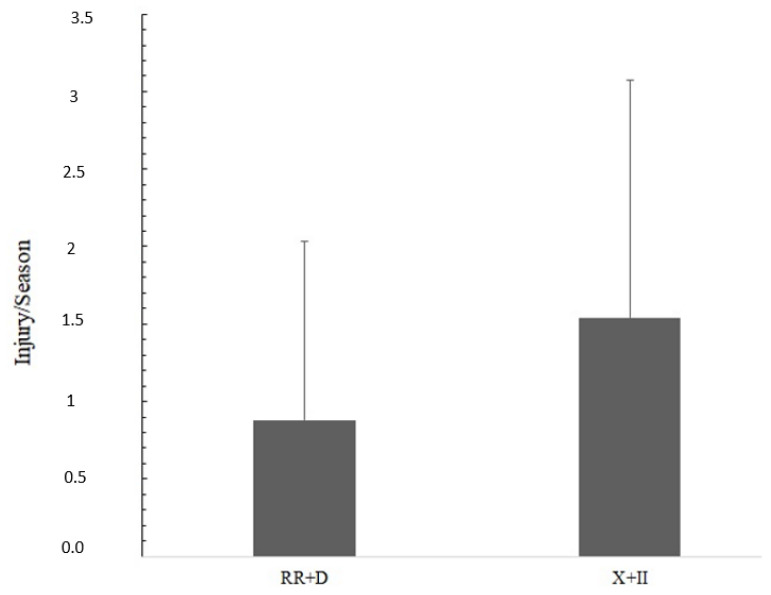
*ACTN3* R577X and *ACE* I/D interaction on the incidence of muscle injury per season, adjusted by weight and age (*p* = 0.016).

**Table 1 genes-13-02009-t001:** Subjects characteristics according to the *ACTN3* R577X and *ACE* I/D genotypes.

	*ACTN3* R577X			*p*-Value *	*ACE* I/D			*p*-Value *
	RR	RX	XX		II	ID	DD	
*n* (%)	34 (40)	38 (45.8)	11 (13.2)		16 (19.3)	34 (41)	33 (39.7)	
Age (years)	24 ± 3	26 ± 5	27 ± 5	0.030	28 ± 5	25 ± 5	24 ± 4	0.094
Height (cm)	179 ± 6	179 ± 6	180 ± 5	0.844	180 ± 6	180 ± 7	179 ± 5	0.733
Weight (kg)	75 ± 6	75 ± 6	72 ± 5	0.270	74 ± 5	75 ± 6	74 ± 5	0.887
Experience(years)	6 ± 4	8 ± 5	9 ± 5	0.079	10 ± 5	7 ± 5	7 ± 5	0.059

Mean ± SD. * The differences between genotypes were evaluated by ANOVA.

**Table 2 genes-13-02009-t002:** Subjects’ characteristics divided between non-severe and severe injury occurrences (presented as the mean ± SD).

	Injury Occurrences		
	Non-Severe	Severe	*p*-Value
*n*	89	10	
Age (years)	26.5 ± 4.9	25.2 ± 4.8	0.426
Height (cm)	180 ± 6	180 ± 4	0.905
Weight (kg)	75 ± 6	71 ± 5	0.032
Experience (years)	8 ± 4	8 ± 5	0.560

**Table 3 genes-13-02009-t003:** Association between the *ACTN3* R577X genotypes and severity and incidence of non-contact muscle injury (per season). The differences between the groups regarding severity were evaluated with the χ2 test, and the differences in injury per season were evaluated with one-way ANOVA. Bold emphasizes values of *p* < 0.05.

		Genotype			Dominant Model		Recessive Model		*p*-Value		
		RR	RX	XX	RR + RX	XX	RR	XX + RX	Genotype	Dominant	Recessive
**Severity (*n*)**	Severe	1	3	6	4	6	1	9	**0.001**	**<0.001**	0.182
	Non-Severe	26	53	10	80	9	26	63			
**Injury/Season**		0.70 ± 1.01	1.18 ± 1.43	1.25 ± 0.86	0.96 ± 1.27	1.25 ± 0.86	0.70 ± 1.00	1.19 ± 1.33	0.213	0.496	0.079

**Table 4 genes-13-02009-t004:** Association between the *ACE* I/D genotypes and severity and incidence of non-contact muscle injury (per season).

		Genotype			Dominant Model		Recessive Model		*p*-Value		
		II	ID	DD	II	ID+DD	II+ID	DD	Genotype	Dominant	Recessive
**Severity (*n*)**	Severe	2	2	6	2	8	4	6	0.154	0.499	0.063
	Non-Severe	29	36	24	29	60	65	24			
**Injury/Season**		1.59 ± 1.51	0.93 ± 1.19	0.77 ± 1.03	1.59 ± 1.51	0.85 ± 1.11	1.12 ± 1.34	0.78 ± 1.01	0.085	**0.03**	0.198

The differences between the groups regarding severity were evaluated with the χ2 test, and the differences in injury per season were evaluated with one-way ANOVA. Bold emphasizes values of *p* < 0.05.

**Table 5 genes-13-02009-t005:** Models of association between the *ACTN3* R577X genotypes (Model 1) and *ACE* I/D (Model 2), and the combination of *ACTN3* R577X and *ACE* I/D (Model 3) with injury severity. Bold emphasizes values of *p* < 0.05.

		*p*-Value	Odds Ratio	95% CI
Model 1	*ACTN3*	**0.010**	5.141	1.472–17.961
	Weight (kg)	0.183	0.907	0.785–1.047
	Experience (years)	0.959	0.996	0.848–1.169
Model 2	*ACE*	0.080	3.437	0.862–13.701
	Weight (kg)	0.066	0.877	0.762–1.009
	Experience (years)	0.818	0.983	0.846–1.141
Model 3	*ACTN3*	**0.011**	4.972	1.445–17.105
	*ACE*	0.081	3.748	0.850–16.533
	Weight (kg)	0.889	0.910	0.776–1.067
	Experience (years)	0.711	1.012	0.858–1.193

## Data Availability

The data presented in this study are available on request from the corresponding author.

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
