# Peer review of "A Pilot Study on the Prediction of Non-Contact Muscle Injuries Based on *ACTN3* R577X and *ACE* I/D Polymorphisms in Professional Soccer Athletes"

_genes, 2022, doi:10.3390/genes13112009_

Round 1

Reviewer 1 Report

Manuscript titles “A pilot study on the prediction of non-contact muscle injuries based on ACTN3 R577X and ACE I/D polymorphisms in professional soccer athletes” adds further knowledge on the association between DNA polymorphisms and muscle injuries among athletes.

The manuscript is well developed, statistical methods are adequate, bibliography is correctly updated.

Here are some suggestions:

1)     In Introduction: please, since it was indicated for ACE, add the rs code also for ACTN3 polymorphism;

2)     In results, page 5: authors show the results of logistic regression with the dominant and recessive model. I wonder why they do not consider also the codominant model. Moreover, I suggest to determine which model is more suitable, for example through Akaike Information Criterion (AIC) or Bayesian Information Criterion (BIC).

3)     In results: authors correctly adjust the analysis taking into account the weight and the age, since in previously analysis they did not find any correlation with years of experience. I would have used another parameter to be evaluated: the volume of training, calculated as hours of exposure to training and matches. I’m not sure if you can get this information from public database, on the contrary it can be indicated as another criticism.

4)     In discussion: authors do not discuss the significant association between genotypes and age and experience years. Is it random?  

5)      In discussion: the part in which authors mention the association between ACTN3 and Duchenne muscular dystrophy is not useful in this context, I suggest to move it in introduction, or eliminate it.

6)     Authors write this is the first study to determine the association of two polymorphisms with the risk of non-contact muscle injury, and this is coherent with the title, but we have other similar studies in literature, as authors reported also in their bibliography, so please better explain the originality of the study.

Author Response

We would like to thank you for Reviewer#1

1)     In Introduction: please, since it was indicated for ACE, add the rs code also for ACTN3 polymorphism; 

====

We added the rs code.

====

2)     In results, page 5: authors show the results of logistic regression with the dominant and recessive model. I wonder why they do not consider also the codominant model. Moreover, I suggest to determine which model is more suitable, for example through Akaike Information Criterion (AIC) or Bayesian Information Criterion (BIC).

====

The AIC pointed out the already chosen models as the most suitable in this case. We reported this analysis and result.

====

3)     In results: authors correctly adjust the analysis taking into account the weight and the age, since in previously analysis they did not find any correlation with years of experience. I would have used another parameter to be evaluated: the volume of training, calculated as hours of exposure to training and matches. I’m not sure if you can get this information from public database, on the contrary it can be indicated as another criticism.

====

The volume of training was not analyzed as this data diverge between the soccer teams, the seasons and the different times of the season. Therefore, this was not analyzed and it was included as one of our limitations.

====

4)     In discussion: authors do not discuss the significant association between genotypes and age and experience years. Is it random?

====

We added a brief discussion regarding the significant association with age. However, our analyses were adjusted by age and weight.

====

5)      In discussion: the part in which authors mention the association between ACTN3 and Duchenne muscular dystrophy is not useful in this context, I suggest to move it in introduction, or eliminate it.

====

We deleted the paragraph.

====

6)     Authors write this is the first study to determine the association of two polymorphisms with the risk of non-contact muscle injury, and this is coherent with the title, but we have other similar studies in literature, as authors reported also in their bibliography, so please better explain the originality of the study.

====

 We highlighted the novelty of the study when it was mentioned and also added a brief explanation in the final paragraph.

=====

Reviewer 2 Report

  • This is an interesting study looking into the combination of genetic polymorphisms (ACTN3 R577X and ACE I/D) that may relate to increase injury incidence and severity. I have some concerns that may need to be addressed. 

    • Given the limited number of severe injuries, results comparing severe injuries across genetic polymorphisms should be interpreted with caution.
    • Looking into the interactions between the ACTN3 and ACE genotypes, I would suggest to also show the other possible interactions in Figure 3. Now it is concluded that those with an ACTN3 X-alle and ACE II genotype have a higher frequency of injuries per season, twice as high than that of other combinations. However, these other combinations are not yet addressed in the results section. 
    • Make sure to use one consistent description of the polymorphisms. So ACE I/D OR ID.
    • Did the authors also consider running a logistic regression using the interaction of both ACE and ACTN3 polymorphisms?

    Specific comments:

    o   Abstract

    §  No background is provided

    §  The aim mentions "alone and in association", please clarify what is meant by this.

    §  Consider rephrasing the sentence starting with "The interaction .."

    §  Consider changing the final sentence into "Our findings suggest that both polymorphisms ACTN3 R577X and ACE I/D (and their interaction) are associated .." and remove the "including in interaction" at the end.

    o   Introduction

    §  Paragraph 1: why "more so in soccer"?

    §  Paragraph 3: do you mean "In this polymorphism, higher strength and power are associated with the R allele, while better muscle endurance is associated with the mutated allele, called the X allele [16,17]"?

    §  Paragraph 4: Consider rephrasing the final sentence.

    §  Paragraph 6: Please explain in more detail what is meant by "with some having conflicting or insufficient results [31-34]".

    §  Paragraph 7: consider changing "athletes" into "soccer players". Also be very clear on what the authors refer to with "association between" (see also my prior comment).

    §  Consider adding the following reference with respect to ACTN3 findings with respect to injury risk/incidence (Front Physiol. 2017 Dec 18;8:1080)

    o   Methods:

    §  Consider rephrasing the sentence starting with "For all the athletes it was collected information on".

    o   Results

    §  Did the results and injury occurrence differ between the 1st and 2nd division soccer players?

    §  Table 1: Please mention in the caption what genotypic comparison the P-values refer to.

    §  The model containing the "ACTN3 R577X genotypes was significant (= 0.004, R2: 0,259)". This R2 should be "0.259"

    §  Table 5: Please also mention the XX genotypes for ACTN3 and DD genotype for ACE in the predictor description of the models. Also, how is injury severity considered in the models, was the outcome severe vs non-severe injuries?

    o   Discussion

    §  Paragraph 2: I would not consider the 2.66 likelihood to be similar than the 5.14 likelihood observed in this study. Consider rephrasing.

    §  What are the practical implications of the present study?

    o   Conclusion

    §  As these findings were observed solely in males, consider adding "male" before "soccer players".

Author Response

This is an interesting study looking into the combination of genetic polymorphisms (ACTN3 R577X and ACE I/D) that may relate to increase injury incidence and severity. I have some concerns that may need to be addressed. 

  • Given the limited number of severe injuries, results comparing severe injuries across genetic polymorphisms should be interpreted with caution.

=====

Considering this, we added this statement as a consequence of our number limitation.  

=====

  • Looking into the interactions between the ACTN3 and ACE genotypes, I would suggest to also show the other possible interactions in Figure 3. Now it is concluded that those with an ACTN3 X-alle and ACE II genotype have a higher frequency of injuries per season, twice as high than that of other combinations. However, these other combinations are not yet addressed in the results section.
  •  

=====

We did not show the other possible combinations as they were not significant. Therefore, we added a sentence in the results section to make it clear.

======

Make sure to use one consistent description of the polymorphisms. So ACE I/D OR ID.

====

We used ACE I/D to refer to the polymorphism and ACE ID genotype to refer to the specific genotype.

=====

  • Did the authors also consider running a logistic regression using the interaction of both ACE and ACTN3 polymorphisms?

=====

We added another model (Model 3) that includes both polymorphisms.

=====

Specific comments:

o   Abstract

  • No background is provided

=====

we added a brief background.

=====

  • The aim mentions "alone and in association", please clarify what is meant by this.

====

The word “alone” was changed by “individually” to make it clear that the polymorphisms were analyzed in isolated but also in association.

====

  • Consider rephrasing the sentence starting with "The interaction .."

====

The phrase was written again for better clarity.

====

  • Consider changing the final sentence into "Our findings suggest that both polymorphisms ACTN3 R577X and ACE I/D (and their interaction) are associated .." and remove the "including in interaction" at the end.

====

We changed accordingly.

====

o   Introduction

  • Paragraph 1: why "more so in soccer"?

====

The explanation for the affirmation was already address din the beginning of the paragraph. We changed “more so in soccer” to another term for clarity.

====

  • Paragraph 3: do you mean "In this polymorphism, higherstrength and power are associated with the R allele, while better muscle endurance is associated with the mutated allele, called the X allele [16,17]"? –

 ====

The words were included for better clarity.

====

  • Paragraph 4: Consider rephrasing the final sentence.

====

The sentence was rephrased for better clarity.

====

  • Paragraph 6: Please explain in more detail what is meant by "with some having conflicting or insufficient results [31-34]".

====

We replaced the mentioned words by examples.

====

  • Paragraph 7: consider changing "athletes" into "soccer players". Also be very clear on what the authors refer to with "association between" (see also my prior comment).

 ====

We specified the athletes as “soccer male athletes”.

====

  • Consider adding the following reference with respect to ACTN3 findings with respect to injury risk/incidence (Front Physiol. 2017 Dec 18;8:1080).

 ====

We added some interesting statements from the reference.

====

o   Methods:

  • Consider rephrasing the sentence starting with "For all the athletes it was collected information on".

====

 Rephrased for better clarity.

====

o   Results

  • Did the results and injury occurrence differ between the 1st and 2nd division soccer players?

====

Because of our limited sample size and also due to the fact that some of the soccer teams had changes in their classification of the Brazilian championship during the analyzed seasons, we did not think it was possible at the moment, but it will be considered for the future and it was also added as suggestion for further studies.

====

  • Table 1: Please mention in the caption what genotypic comparison the P-values refer to.

====

We added that it is related to the differences between genotypes and evaluated by ANOVA.

====

  • The model containing the "ACTN3 R577X genotypes was significant (= 0.004, R2: 0,259)". This R2 should be "0.259"

=====

– Changed to 0.259

=====

  • Table 5: Please also mention the XX genotypes for ACTN3 and DD genotype for ACE in the predictor description of the models. Also, how is injury severity considered in the models, was the outcome severe vs non-severe injuries?
  •  

====

– Added the description clarifying that the comparison is the outcome severe vs non-severe and added the reference of XX and DD genotypes as predictors in the models.

====

o   Discussion

  • Paragraph 2: I would not consider the 2.66 likelihood to be similar than the 5.14 likelihood observed in this study. Consider rephrasing.

====

We have rephrased.

===

  • What are the practical implications of the present study?
  •  

=====

A new paragraph was added to mention the implications of the study.

=====

o   Conclusion

  • As these findings were observed solely in males, consider adding "male" before "soccer players".

=====

Added the term.

====

Round 2

Reviewer 2 Report

N/A